# PD-L1^+^ Lymphocytes Are Associated with CD4^+^, Foxp3^+^CD4^+^, IL17^+^CD4^+^ T Cells and Subtypes of Macrophages in Resected Early-Stage Non-Small Cell Lung Cancer

**DOI:** 10.3390/ijms251910827

**Published:** 2024-10-09

**Authors:** Giedrė Gurevičienė, Jurgita Matulionė, Lina Poškienė, Skaidrius Miliauskas, Marius Žemaitis

**Affiliations:** 1Department of Pulmonology, Medical Academy, Lithuanian University of Health Sciences, LT-44307 Kaunas, Lithuania; 2Department of Pathology, Medical Academy, Lithuanian University of Health Sciences, LT-44307 Kaunas, Lithuania

**Keywords:** programmed death-ligand 1, tumour microenvironment, lymphocytes, macrophages, non-small cell lung cancer

## Abstract

The non-canonical PD-L1 pathway revealed that programmed-death ligand 1 (PD-L1) expression in immune cells also plays a crucial role in immune response. Moreover, immune cell distribution in a tumour microenvironment (TME) is pivotal for tumour genesis. However, the results remain controversial and further research is needed. Distribution of PD-L1-positive (PD-L1^+^) tumour-infiltrating lymphocytes in the context of TME was assessed in 72 archival I–III stage surgically resected NSCLC tumour specimens. Predominant PD-L1^+^ lymphocyte distribution in the tumour stroma, compared to islets, was found (*p* = 0.01). Higher PD-L1^+^ lymphocyte infiltration was detected in smokers due to their predominance in the stroma. High PD-L1^+^ lymphocyte infiltration in tumour stroma was more common in tumours with higher CD4^+^ T cell infiltration in islets and stroma, Foxp3^+^CD4^+^ T cell infiltration in islets and lover M1 macrophage infiltration in the stroma (*p* = 0.034, *p* = 0.034, *p* = 0.005 and *p* = 0.034 respectively). Meanwhile, high PD-L1^+^ lymphocyte infiltration in islets was predominantly found in tumours with high levels of IL-17A^+^CD4^+^ T cells in islets and Foxp3^+^CD4^+^ T cells in islets and stroma (*p* = 0.032, *p* = 0.009 and *p* = 0.034, respectively). Significant correlations between PD-L1^+^ lymphocytes and tumour-infiltrating CD4^+^, Foxp3^+^CD4^+^, IL-17A^+^CD4^+^ T cells and M2 macrophages were found. An analysis of the tumour-immune phenotype revealed a significant association between PD-L1 expression and IL17^+^CD4^+^ and Foxp3^+^CD4^+^ immune phenotypes. PD-L1^+^ lymphocytes are associated with the distribution of CD4^+^, Foxp3^+^CD4^+^, IL17A^+^CD4^+^ T cells, M1 and M2 macrophages in TME of resected NSCLC.

## 1. Introduction

Lung cancer remains one of the most prevalent oncological diseases and the leading cause of cancer death worldwide [1]. The discovery of immune checkpoints such as programmed death 1 (PD-1) and programmed death ligand 1 (PD-L1) has made a breakthrough in cancer immunology and therapy in a wide variety of cancer types, including non-small cell lung cancer (NSCLC). Blocking the PD-1/PD-L1 pathway with immunotherapy has been shown to be beneficial in advanced, locally advanced and, recently, in early resected NSCLC patients. However, the benefit of immune checkpoint inhibitors remains modest. Despite the fact that a positive correlation between PD-L1 expression and a favourable prognosis was determined, PD-L1 is not a perfect predictive biomarker in advanced or early-stage NSCLC [2,3,4,5,6,7,8]. These contrasting results could be due to several diagnostic immunohistochemistry (IHC) assays with different scoring systems, approved for PD-L1 testing in NSCLC, different patient populations, but also can be determined by tumour microenvironment (TME), in which three main immune phenotypes according the distribution of immune cells can be identified: immune “inflamed”, which contain high levels of T cells in the intra-tumoural compartment; immune “excluded”, which contain T cell infiltration in tumour stroma, but not tumour islets; and “desert”, which exhibit no immune infiltrates [9,10,11]. The results of research evaluating the association between immune phenotypes and clinical outcomes remain controversial. Moreover, the mechanisms responsible for immune cells’ special distribution in tumour tissue are largely unclear [12,13]. It has been hypothesised that not only T cells but the entire TME, including macrophages and other immune and non-immune cells, are of exceptional importance [14,15]. Furthermore, besides the canonical, the non-canonical PD-1/PD-L1 axis function mode was revealed. It was found that tumour-infiltrating PD-L1^+^ lymphocytes can interact with PD-1^+^ cells, such as other tumour-infiltrating lymphocytes or macrophages, resulting in bidirectional signalling and immunosuppressive TME, including the inhibition of CD4^+^ T cells and CD8^+^ T cells activity, inducing M2 polarization of macrophages and the differentiation of Tregs and Th17, resulting in enhanced secretion of pro-tumourigenic immunosuppressive factors, such as TGF-β, IL-10 and IL-17 [15,16].

In addition, a variety of studies have already demonstrated that tumour-infiltrating PD-L1^+^ lymphocyte levels have the potential to serve as a prognostic factor, but the results remain controversial [12,13]. However, these studies focused only on PD-L1^+^ tumour-infiltrating immune cells’ existence in tumour tissue without taking into consideration these cell distributions in tumour islets and stroma. While there are no data about PD-L1^+^ lymphocyte distribution impact on prognosis in NSCLC, the research in ovarian carcinoma revealed that neither PD-L1^+^ lymphocytes in the tumour stroma nor PD-L1^+^ lymphocytes in tumour islets had a significant association with prognostic variables [17]. Meanwhile, a series of previous studies have demonstrated that precise immune cell distribution in TME is a crucial factor for immune response formation, dictating therapeutic response and identifying prognostic biomarkers [18,19,20]. To our knowledge, we are the first to investigate PD-L1^+^ lymphocyte distribution in the context of the tumour-immune phenotype in NSCLC.

The aim of this study was to evaluate the value of PD-L1^+^ lymphocyte distribution in TME in the context of the tumour-immune phenotype in early-stage resected NSCLC patients.

## 2. Results

The PD-L1 expression in TME was assessed in 72 resected early-stage NSCLC specimens. The clinicopathological characteristics of the patients were already published in the previous article and are reproduced in Table 1 [21].

In our previous study, we found that PD-L1 expression evaluated by TPS, IC, and CPS was significantly associated with the CD4^+^ T cells, Foxp3^+^CD4^+^ T cells, CD8^+^ T cells and M2 macrophage infiltration in the whole tumour tissue [21]. Meanwhile, other studies have demonstrated not only the existence of tumour-infiltrating immune cells but also that their distribution in TME (islets and stroma) may serve to predict the outcomes as well as their response to therapy [18,19,22]. Therefore, we proceeded with our investigation and evaluated the association between PD-L1 TPS, IC, and CPS, as well as the immune cell infiltration level and the distribution in TME. Although PD-L1 expression did not differ depending on CD4^+^ T cells, CD8^+^ T cells or macrophage distribution in TME, a significant association between PD-L1 expression and subsets of CD4^+^ T cells: Foxp3^+^CD4^+^ T cells as well as IL-17A^+^CD4^+^ T cell distribution in TME has been found. Tumours with positive PD-L1 IC expression were more likely to have high Foxp3^+^CD4^+^ T cell infiltration in tumour islets, as well as tumours with positive PD-L1 TPS, IC, or CPS were more likely to have low IL-17A^+^CD4^+^ T cell infiltration in the tumour stroma (Figure 1A–D, Appendix A).

Since PD-L1 TPS, IC, and CPS indexes are described as the proportion of the whole tumour area occupied by PD-L1-positive tumour or immune cells regardless of department of these cells, we further analyzed specifically PD-L1^+^ lymphocyte distribution in TME (tumour islets and stroma) in 10 most representative high-power fields (HPFs ×400 magnification) per tissue section in all 72 cases, including those with negative PD-L1 expression evaluated by TPS, IC, and CPS. Predominant infiltration of PD-L1^+^ lymphocytes was observed in tumour stroma compared with tumour islets (median 36.5 vs. 5, *p* = 0.01). PD-L1^+^ lymphocyte distribution in TME, depending on clinicopathological features, is presented in Table 2. Significantly higher PD-L1^+^ lymphocyte infiltration in tumour islets was observed in older patients. Furthermore, the former or current smokers had significantly higher PD-L1^+^ lymphocyte infiltration due to the predominance of these cells in the stroma compared to the non-smokers. Although the higher PD-L1^+^ lymphocyte infiltration was also detected in men, patients with COPD and squamous cell histology, a significant difference was not reached (Table 2).

A significant correlation was found between smoking intensity, evaluated as pack-years, and PD-L1^+^ lymphocyte infiltration irrespective of their distribution in TME (r = 0.284, *p* = 0.021 in tumour stroma; r = 0.294, *p* = 0.016 in tumour islets; r = 0.276, *p* = 0.025 in total; respectively) as well.

Furthermore, we analyzed the association between the infiltration levels of PD-L1^+^ lymphocytes and tumour-infiltrating immune cells depending on the distribution of these cells in TME. We detected significant associations between the infiltration levels of PD-L1^+^ lymphocytes and CD4^+^ T cells, Foxp3^+^CD4^+^ T cells, IL-17A^+^CD4^+^ T cells, as well as M1 macrophages in different compartments of TME (Table 3). Higher infiltration of PD-L1^+^ lymphocytes in tumour stroma was associated with higher infiltration levels of CD4^+^ T cells not only in stroma but in tumour islets as well. The same but not significant tendency was detected for PD-L1^+^ lymphocytes in tumour islets. Further analysis of the subsets of CD4^+^ T cells showed that tumours with higher infiltration levels of PD-L1^+^ lymphocytes in tumour islets were significantly associated with higher IL-17A^+^CD4^+^ T cell infiltration in the same compartment. A similar pattern was observed for PD-L1^+^ lymphocytes in tumour stroma. Furthermore, tumours with higher infiltration levels of PD-L1^+^ lymphocytes in tumour islets were significantly associated with higher Foxp3^+^CD4^+^ T cell infiltration, despite the distribution of these cells. On the other hand, higher PD-L1^+^ lymphocyte infiltration in tumour stroma was associated only with higher Foxp3^+^CD4^+^ T cell infiltration in islets and lower M1 macrophage infiltration level in tumour stroma (Table 3).

Significant correlations (from week to moderate) between PD-L1^+^ lymphocytes and CD4^+^ T cells, IL-17A^+^CD4^+^ T cells, Foxp3^+^CD4^+^ T cells and M2 macrophage distributed in different TME compartments were detected (Figure 2A–H, Appendix A).

We further investigated whether PD-L1 expression could depend on the tumour-immune phenotype and immune cell distribution in TME in resected NSCLC. Firstly, we grouped tumours into immune “desert”, “excluded” and “inflamed” immune phenotypes for different immune cells; the results are represented in Figure 3. The most frequent immune phenotype was „inflamed” and accounted for half of the cases for most of the tumours. IL17A^+^CD4^+^ T cells demonstrated the highest proportion of „inflamed“ immune phenotype and accounted for almost two-thirds of the cases. The remaining number of tumours split into quite equal parts for “desert” and “excluded” immune phenotypes, except for Foxp3^+^CD4^+^ and M1 tumours, where “desert” was more common than the “excluded” immune phenotype (Figure 3).

Subsequently, we analysed the association between the tumour-immune phenotypes and PD-L1 expression, evaluated by TPS, IC, and CPS. PD-L1 TPS and CPS were significantly associated with the IL17A^+^CD4^+^-immune phenotype—as well as PD-L1 IC and CPS—with the Foxp3^+^CD4^+^ immune phenotype. The highest prevalence of PD-L1 IC or CPC ≥ 10% was detected in the Foxp3^+^CD4^+^ “inflamed” immune phenotype, compared to other immune phenotypes, while PD-L1 CPC ≥ 10% was totally absent in IL17A^+^CD4^+^ “excluded” immune phenotype. The same results were detected for IL17A^+^CD4^+^ “excluded” immune phenotype, and PD-L1 TPS ≥ 1%. (Figure 4A,B, Appendix A; Figure 4C,D, Appendix A).

We also evaluated the association between the infiltration level and distribution of PD-L1^+^ lymphocytes and immune phenotype. PD-L1^+^ lymphocyte infiltration level was significantly associated with the Foxp3^+^CD4^+^ immune phenotypes regardless of the compartment—the highest in the “inflamed” immune phenotype and the lowest in the “desert” immune phenotype with intermediate results in the “excluded” immune phenotype. This repeated the tendencies that were seen between CD4^+^ phenotypes and PD-L1^+^ lymphocyte infiltration; however, the level of significance was not reached. Analysing the IL17A^+^CD4^+^-immune phenotypes, only the “excluded” immune phenotype had the lower level of PD-L1^+^ lymphocyte infiltration, with very similar results for the other immune phenotypes. This, again, repeated the trend that had already been seen when evaluating the association between CD8^+^ immune phenotypes and tumour stroma infiltrating PD-L1^+^ lymphocytes without statistical significance (Table 4).

However, there were no significant associations between M1 or M2 immune phenotypes and PD-L1^+^ lymphocyte distribution in tumour tissue (Table 5).

The numbers of PD-L1^+^ lymphocytes in tumour islets and stroma, depending on the immune phenotype, are represented as median with range in Appendix A. These results have repeated the tendencies that were seen in evaluating the association between infiltration level and distribution of PD-L1^+^ lymphocytes and immune phenotype.

However, there was no significant association between PD-L1^+^ lymphocyte distribution in tumour tissue and survival, neither alone, nor in the context of TME.

## 3. Discussion

The goal of this study was to deepen the understanding of TME, evaluating the spatial distribution of tumour-infiltrating immune cells, adding special emphasis on PD-L1^+^ lymphocytes and their association with the tumour-immune phenotype, as well as PD-L1 expression and prognosis. To our knowledge, we are the first who evaluated PD-L1^+^ lymphocyte distribution in tumour tissue in the context of the TME (CD4^+^ T cells, Foxp3^+^CD4^+^ T cells, IL17A^+^CD4^+^ T cells, CD8^+^ T cells, M1 and M2 macrophages).

The discovery of a non-canonical PD-L1 pathway led to the exploration of PD-L1 expression in terms of tumour-infiltrating immune cells and their distribution in tumour tissue. Therefore, we analysed PD-L1^+^ lymphocyte distribution in TME and found predominant PD-L1^+^ lymphocyte infiltration in the tumour stroma, compared to the tumour islets. It can be assumed that immunosuppressive microenvironment driven by PD-L1^+^ lymphocytes predominate in tumour stroma [23,24]. Other studies using formalin-fixed specimens from human lung tumours have also demonstrated that T cells accumulate more frequently in the stroma compared to the tumour islets. Salmon et al. hypothesised that this could be due to aligned fibres around tumour epithelial cells and in perivascular regions. These aligned fibres could dictate the trajectory of the T-cell migration and restrict them from stepping into tumour islets [23].

However, Wu et al. demonstrated in an NSCLC study that tumours with a high infiltrating PD-L1^+^ lymphocyte level in the stroma were more frequently detected with high CD8^+^ T effector cells, demonstrating positive PD-1 expression [25]; we have found that higher infiltration of CD4^+^ T cells but not CD8^+^ T cells was associated with higher PD-L1 expression on lymphocytes irrespective of their distribution in TME. Wu et al. included not only the early stages but also advanced and metastatic cancer, and the association between PD-L1^+^ lymphocytes with other T cells was not performed [25]. It is well known that CD8^+^ T lymphocytes act as tumour killers after the activation under immunotherapy. On the other hand, CD4^+^ T cells can promote immunosuppressive behaviour in TME. Already two decades ago, it was discovered that PD-L1^+^ T cells can interact with PD-1 on other T-cells and reduce T-cell proliferation, as well as IL-2 and IFN-γ production via the alloreactive T–T interaction [26]. Latchman et al. in mice model revealed that PD-L1 on T cells negatively regulates these cell effects: PD-L1^−/−^ mice demonstrated markedly enhanced CD4^+^ and CD8^+^ T cell responses, compared with wild-type mice [27]. Moreover, in other mice model studies, Tamura et al. reported that the costimulatory effect of PD-L1 was more favourable for CD4^+^ T cells than for CD8^+^ T cells. When purified CD4^+^ and CD8^+^ T cells were stimulated with the same concentration of anti-CD3 and mB7-H1Ig, 10-fold higher proliferation of CD4^+^ T cells was seen, whereas CD8^+^ T cells proliferation improved only 2- to 3-fold [28]. These early findings confirmed that PD-L1^+^ T cells, especially CD4^+^ T cells, may act as a receptors and receive signals that affect T-cell function.

Further analysis of the CD4^+^ T lymphocyte subtypes revealed an association between PD-L1 expression and Foxp3^+^CD4^+^ T cells, as well as IL-17A^+^CD4^+^ T cells distributed in the TME. Tumours with positive PD-L1 IC expression were more likely to have high Foxp3^+^CD4^+^ T cell infiltration in tumour islets. The same tendency has been observed in analyzing PD-L1^+^ lymphocytes. Higher PD-L1^+^ lymphocyte infiltration, regardless of the compartment, was associated mostly with higher Foxp3^+^CD4^+^ T cell infiltration in islets. While Foxp3^+^CD4^+^ T cells, also known as Tregs, distribution in tumour tissue alone has been extensively studied, the results remain controversial. While the bulk of solid tumour studies, including NSCLC, demonstrated a potentially worse prognosis for the patients with high tumour-infiltrating Foxp3^+^CD4^+^ T cell levels, others, to the contrary, represented results with better prognoses for patients with a high level of tumour stroma infiltrating Foxp3^ +^CD4^+^ T cells. These results lead to the hypothesis that these cells might play a dual role in carcinogenesis [22]. It is known that Tregs demonstrate pro-tumourigenic effects in an IL-10-dependent manner. Shiri et al., in a mouse model, revealed that IL-10 acted on Tregs in an autocrine manner, leading to increased IL-10 production. Moreover, IL-10 acted on monocytes as well as other myeloid cells and induced the upregulation of the PD-L1. Finally, the PD-L1/PD-1 pathway suppressed CD8^+^ T cell-dependent cytotoxicity against metastatic lesions in the liver [29]. Moreover, it was found that immune-desert or immune-excluded tumours, which are resistant to anti-PD-1/PD-L1 therapy, could benefit from anti-TGF-β/PD-L1 combined therapy [30]. 

While it is known that TGF-β can be secreted by Tregs, the effect of TGF-β on PD-L1 expression remains unknown. Further analysis revealed that PD-1^+^ Tregs modulate VE-cadherin through PI3K/Akt and ERKs pathways and regulate VCAM-1 expression in lymphatic endothelial cells (LEC) through the classical Nuclear factor-κB (NF-κB) p65 pathway, leading to the regulation of Tregs and T-effector cells trans-endothelial migration [31]. However, the precise mechanism of how exactly PD-L1^+^ Tregs can be associated with these cells’ migration to tumour islets remains unknown, and further investigation is required. Based on this information, it can be assumed that the immunosuppressive function of Foxp3^+^CD4^+^ T cells in tumour tissue can be driven by PD-L1 expression in these cells localised in tumour islets.

Analyzing IL-17A^+^CD4^+^ T cells, the results are more contradictory. Tumours with positive PD-L1 TPS, IC, or CPS expression were more likely to have low IL-17A^+^CD4^+^ T cell infiltration in the tumour stroma. On the other hand, tumours with a higher infiltration level of PD-L1^+^ lymphocytes in tumour islets or stroma were significantly associated with higher IL-17A^+^CD4^+^ T cell infiltration in tumour islets. Meanwhile, IL17A^+^CD4^+^ T cells in tumour tissue were of less interest, and there is no data representing the effect of these cells, depending on their compartment of tumour tissue. However, similar to Foxp3^+^CD4^+^ T cells data, the presence of IL17A^+^CD4^+^ T cells in the tumour microenvironment is associated with potential anti-tumour, as well as pro-tumourigenic effects due to their plasticity [32]. According to the medical data, CD4^+^ T cell subpopulation Th17 is generally associated with a pro-tumourigenic immune response gained through cell-to-cell contact with anti-inflammatory cytokines IL-10 in a TGF-β-dependent manner, which induces other immune cell population suppression. 

However, recent studies also demonstrate that an IFN-γ-dependent manner can lead to the recruitment of effector cells into the TME and also induce an anti-tumour response. Moreover, Th17 can also demonstrate direct cytotoxic response through TNF and IFN-γ secretion, the production of granules with a cytotoxic effect, or the expression of a tumour necrosis factor (TNF)-related apoptosis-inducing ligand (TRAIL) that after engagement with receptors, induces cancer cell apoptosis [33,34]. PD-L1 engagement on T-cells suppresses the differentiation of Th1 cells and promotes the differentiation of Th17 cells via the STAT3 pathway, inducing an anergic phenotype in CD8^+^ T cells [16]. Our results may suggest that IL17A^+^CD4^+^ T cells associated with PD-L1^+^ lymphocytes can lead to immunosuppression and a tumour-immune escape acting in tumour islets.

Further analysis revealed a significant association between tumour-infiltrating PD-L1^+^ lymphocytes and macrophages. Higher PD-L1^+^ lymphocyte infiltration in tumour stroma was associated with lower M1 macrophage infiltration levels in the same compartment. Whereas a significant correlation between PD-L1^+^ lymphocytes in tumour islets and M2 macrophages distributed in tumour stroma was detected. To our knowledge, we are the first ones to represent the associations between PD-L1^+^ lymphocyte distribution in tumour tissue and M1 or M2 macrophages. According to the medical data, tumour-infiltrating PD-L1^+^ T cells can act on PD-1^+^ macrophages and suppress M1 polarization by reducing NF-κB and signal transducer, as well as activator of transcription (STAT) 1 phosphorylation, but induce M2 polarization by increasing STAT6 phosphorylation. Moreover, clinical studies with anti-PD-1 therapy have demonstrated that promoted M1 polarization may directly function on PD-1^+^ macrophages [16]. Our results further support the theory that M1 macrophage polarization is suppressed and M2 polarization is promoted predominantly in tumour stroma by the PD-L1^+^ lymphocytes that bind to PD-1^+^ cells in the TME.

Tumours historically are divided into phenotypes according to CD8^+^ T cell distribution, while CD8^+^ T cells are known to be the most powerful effectors in the anticancer immune response or according to total lymphocyte distribution in tumour tissue [35]. However, as described earlier, other T cell subtypes, as well as macrophages, also play an indisputably important role in tumour genesis. Therefore, we grouped the tumours into immune “desert”, “excluded”, and “inflamed” immune phenotypes for all different immune cells.

The PD-L1^+^ lymphocyte infiltration level was the highest in the Foxp3^+^CD4^+^ T cell “inflamed” immune phenotype, the lowest in the “desert” immune phenotype, and intermediate in the “excluded” immune phenotype. The highest probability of detecting PD-L1 IC and CPS ≥ 10% was in the Foxp3^+^CD4^+^ “inflamed” phenotype. The same pattern was seen for the CD4^+^ but not CD8^+^ immune phenotypes.

According to the clinical data, the “inflamed” tumour-immune phenotype is associated with a worse prognosis but with favourable outcomes in patients receiving immunotherapy [20]. Recent animal model studies were used to help identify immunological mechanisms leading to the therapeutic response. Tilsed et al. used RNA sequencing in an animal model, revealing that in response to chemotherapy, tumour-infiltrating CD4^+^ T cells play a crucial role in breast cancer. Tumours enriched with inflammatory genes, particularly in IFN- and TNF-α-related pathways and CD4^+^ T cells in an inflamed tumour microenvironment, were associated with better response to chemotherapy. This is in line with the results of immunotherapy: a better response is demonstrated in patients with the “inflamed” immune phenotype [36]. Furthermore, Zander et al. demonstrated in a “desert” immune phenotype mice model that CD4^+^ T cells, in an IL-21-dependent manner, can induce CD8^+^ T cell differentiation into protective CX_3_CR1^+^ CD8^+^ T cells, leading to a more than two-fold increase in these cells in the TME [37]. Unfortunately, the mechanism how exactly the Foxp3^+^CD4^+^ T cells expressing PD-L1 act in the TME remains unclear. However, our findings, together with other scientific data, suggest that PD-L1^+^ lymphocytes are more often detected in the Foxp3^+^CD4^+ “^inflamed” phenotype, and these factors altogether could provide additional information regarding patient selection for immune or chemotherapy. However, further investigation is required.

Moreover, PD-L1 TPS and CPS were significantly associated with the IL17A^+^CD4^+^-immune phenotype. The IL17A^+^CD4^+^ “excluded” immune phenotype had a lower level of PD-L1^+^ lymphocyte infiltration, while the infiltration levels in “inflamed” and “desert” immunophenotypes were fairly evenly distributed. The same tendencies were seen when the association between a number of PD-L1^+^ lymphocytes distributed in tumour islets and stroma and the IL17A^+^CD4^+^-immune phenotype was assessed. Unfortunately, the information regarding PD-L1^+^ IL17A^+^CD4^+^ T cells is insufficient, and the mechanism of our findings remains unknown. Interestingly, evaluating the association between CD8^+^ immune phenotypes and tumour stroma, infiltrating PD-L1^+^ lymphocytes, repeated the trends seen in the IL17A^+^CD4^+^-immune phenotype, but the statistical significance was not reached. The link between these two phenotypes could be IL17, which is crucial in IL17A^+^CD4^+^ T cell activity but also plays an important role in CD8^+^ T cell activity. In multiple sclerosis, CD8^+^ T cells producing IL17 are enriched in active lesions, suggesting an important role of these cells in the pathogenesis of autoimmunity [38]. Furthermore, the IL-17 signalling deficiencies in the hosts can result in decreased CD8^+^ T cell anti-tumour immunity and cause tumour-specific changes in the lymphoid cell populations [39]. Further research is required to deepen the understanding of these findings.

In our previous study, we reported that PD-L1 expression, evaluated by three scoring methods (TPS, IC, and CPS), correlated with smoking intensity, defined as pack years [21]. In this study, we have also found a significant correlation between PD-L1^+^ lymphocyte infiltration, irrespective of their compartment and smoking intensity. Moreover, significantly higher PD-L1^+^ lymphocyte infiltration in tumour tissue in current or former smokers, compared to non-smokers, was observed, and this was due to this cell distribution in the tumour stroma. This is in agreement with Song et al., who evaluated histological tumour specimens for PD-L1 expression on tumour and immune cells separately in 305 patients diagnosed with stage I–IV NSCLC and found that smoking was associated not only with PD-L1^+^ expression in tumour cells but also in immune cells [40]. Meanwhile, head and neck squamous cell carcinoma had significantly lower PD-L1^+^-immune cell infiltration in tumour islets, and a tumour margin was observed in current smokers compared to never-smokers, whereas there was no association between the PD-L1^+^-immune cells in stroma and smoking [41]. Nevertheless, the weight of evidence demonstrates that smoking tobacco has a diverse effect, including immunosuppressive and pro-inflammatory. However, the exact mechanism of how tobacco smoke affects the PD-L1^+^ lymphocytes remains unknown, and further investigation is needed [42].

In our study, we found that PD-L1^+^ lymphocytes are not a reliable predictive factor for prognosis in resected NSCLC. This is in line with other research involving NSCLC and other solid tumours, demonstrating that PD-L1^+^ lymphocytes are not associated with the disease prognosis. A meta-analysis of 15 studies, including urinary tract, soft tissue, breast, ovary, pancreatic, gastrointestinal tract, and lung carcinomas involving 7251 patients, evaluated the relationship between PD-1^+^ lymphocytes, PD-L1^+^ lymphocytes, and prognosis, but the results were controversial. Interestingly, the PD-L1^+^ lymphocytes had a positive impact on overall survival in studies using a tissue microarray but had a minimal impact when only whole tissue sections were considered and were predictive, for example, for gastrointestinal tract carcinomas. However, the subgroup analysis revealed that NSCLC case only PD-1^+^ lymphocytes but not PD-L1^+^ lymphocytes were associated with better overall survival and disease-free survival results [43]. The same tendency was found in a study where 197 specimens of FIGO stage I–IV ovarian cancer were evaluated for PD-1^+^ and PD-L1^+^ tumour-infiltrating immune cells and prognosis: only the PD-1^+^ immune cells were associated with better OS results. 

Interestingly, when PD-1^+^ and PD-L1^+^ immune cells were evaluated, taking into account their distribution in tumour tissue, neither these cells’ infiltration in tumour islets nor in tumour stroma demonstrated any significant correlation with prognostic variables; however, it is important to mention that in this case, the evaluated immune cells included not only lymphocytes but granulocytes and macrophages as well [17]. These controversial results could be due to a number of reasons. First of all, PD-L1 evaluation only on lymphocytes is not usually used in daily clinical practice, and the stratification of the evaluation is needed. Moreover, as described in our previous article, when PD-L1 expression was evaluated using three scoring methods, different studies, just like in this case, performed immunohistochemistry using several assays with diverse thresholds for positivity, variation of tumour sample size (biopsy vs. resection specimens), and localization (primary tumour vs. metastasis), which could also lead to controversial results [44,45].

The main limitations of this study are its small sample size and the use of archival histological specimens. While this was a continuation of our previous work and the assessment of tumour-infiltrating immune cells was already conducted, we also had not performed a double-staining, which might allow for the PD-L1 expression to be evaluated directly on specific immune cell subtypes. Further studies of PD-L1^+^ lymphocytes are required within the context of the TME in early-stage NSCLC.

## 4. Materials and Methods

### 4.1. Methods

This retrospective study was approved by the Kaunas Regional Ethics Committee for Biomedical Research (No. BE-2-44).

### 4.2. Study Population

A total of 72 patients who were diagnosed with stage I–III NSCLC between September 2012 and February 2015 and underwent surgical resection at the Hospital of Lithuanian University of Health Sciences, Kaunas Clinics, were included. All lung tumour samples were taken from surgical excision specimens. Evaluation of tumour-infiltrating immune cells (tumour-infiltrating lymphocytes (TILs) and tumour-infiltrating macrophages (TAMs)), as well as PD-L1 expression evaluated as TPS, IC, and CPS were already conducted in prior studies [18,19,21,22].

The 7th edition of the TNM Classification of Malignant Tumours was used for lung tumour staging [46]. The histology was categorised according to the World Health Organization (WHO) classification of lung tumours [47]. None of the patients had received neoadjuvant therapy, but adjuvant chemotherapy or radiotherapy may have been administered for patients with stage II/III disease. Patients’ pathological and clinical information was retrospectively obtained from biopsy reports and medical records.

A history of other cancers, connective tissue disorders, unstable systemic conditions (such as severe cardiovascular disease or persistent infections), as well as exacerbation of COPD and systemic glucocorticoid therapy in the month prior to surgery, were all considered as exclusion criteria.

Smoking status was determined by the number of cigarettes smoked during the lifetime. Those patients who had smoked more than 100 cigarettes were classified as smokers (either present or past); those who had not—were classified as non-smokers. Pack-years were calculated by multiplying the number of packs of cigarettes the patient smoked each day by the number of years and were used for quantification of smoking intensity.

### 4.3. Immunohistochemistry Analysis

All blocks used for immunohistochemistry (IHC) staining were taken from surgical excision specimens. Formalin-fixed, paraffin-embedded (FFPE) tissue of NSCLC was sectioned into 3–5 µm thick slices. The IHC for PD-L1 was performed using mouse monoclonal anti-PD-L1 antibody, clone 22C3 pharmDx (Agilent Technologies/Dako, Carpinteria, CA, USA), on the Roche Ventana Benchmark XT automated slide stainer (Ventana Medical Systems, Roche, Neuilly sur Seine, France), following manufacturer’s instructions. For each slide, positive control of the placenta and human tonsils were used. PD-L1 IHC was evaluated by two investigators under a light microscope (Olympus BX50 microscope (Olympus Co., Tokyo, Japan)). For difficult cases or disagreements, a consensus has been made after reviewing and discussing the specimen.

Scoring of tumour-infiltrating immune cells (CD8^+^ T cells, CD4^+^ T cells, Foxp3^+^CD4^+^ T cells, interleukin (IL)-17A^+^CD4^+^ T cells, M1 and M2 macrophages), as well as PD-L1 expression (evaluated by TPS, IC, and CPS) has been previously reported [18,19,21,22]. Briefly, quantitative evaluation of tumour-infiltrating immune cells was performed in the 10 most representative high-power fields (DPFs ×400 magnification) per tissue section. PD-L1 expression was evaluated by TPS (the number of viable tumour cells expressing PD-L1 (complete or partial circumferential linear staining) at any intensity, divided by all viable tumour cells in the examined section and multiplied by 100), IC (the proportion of tumour area occupied by PD-L1-positive lymphocytes and macrophages with PD-L1 staining (membrane and/or cytoplasmic) at any intensity) and CPS (the sum of PD-L1-positive tumour cells and immune cells (lymphocytes and macrophages) divided by the total number of viable tumour cells and multiplied by 100) was performed in the whole tissue section.

While previous studies have reported some significant association between the tumour-immune phenotype, described by tumour-infiltrating immune cell distributions in tumour tissue, and PD-L1 expression with implication to prognosis, we grouped tumours into three subgroups according to high and low immune cell infiltration, taking median as a threshold. “Desert” immune phenotype was described as low immune cell infiltration in tumour islets and stroma. “Excluded” immune phenotype was determined as high immune cell infiltration in tumour stroma but low infiltration in tumour islets. Tumours with high immune cell infiltration in tumour islets were considered to be an “inflamed” immune phenotype. In line with this, PD-L1 expression evaluated by TPS, IC, and CPS was grouped into: negative (TPS, IC, and CPS < 1%) and positive (TPS, IC, and CPS ≥ 1%) subgroups; low (TPS < 50%, IC < 10%, CPS < 10%) and high (TPS ≥ 50%, IC ≥ 10%, CPS ≥ 10%) subgroups. Further evaluation of the association between the tumour-immune phenotypes and PD-L1 expression subgroups and a prognosis was performed.

Further evaluation of PD-L1 expression on tumour-infiltrating lymphocytes was assessed. In order to evaluate staining quality and preservation, slides stained with hematoxylin and eosin were assessed first. Subsequently, the control group of human tonsils and placenta were examined. Finally, quantitative PD-L1 expression on lymphocytes was performed in 10 most representative high-power fields (HPFs ×400 magnification) per tissue section. Only slides with at least 100 viable tumour cells were considered suitable for evaluation. Necrotic and degenerated tumour cells were excluded from the PD-L1 expression evaluation. The number of PD-L1^+^ lymphocytes in NSCLC was counted manually in two compartments: tumour islets and tumour stroma. PD-L1 staining is represented in Figure 5. The number of total PD-L1^+^ lymphocytes was expressed as the sum of PD-L1^+^ lymphocytes in tumour islets and stroma. Lymphocytes at any intensity of membrane and/or cytoplasmic PD-L1 staining were considered positive. Furthermore, the level of PD-L1^+^ lymphocytes was categorised into two subgroups (low and high) based on cell count values below and above the median.

### 4.4. Statistical Analysis

The statistical analysis was performed using Statistical Package for the Social Sciences (SPSS), version 25. The Kolmogorov–Smirnov test was used to assess the sample normality. For non-normally distributed data, a median containing the minimum and maximum values was used. The differences between the two independent variables were estimated using the Mann–Whitney U test. Meanwhile, the Kruskal–Wallis test was used to assess the differences between more than two independent variables. Categorical data analysis was performed using the Chi-square (χ^2^) test or Fisher’s exact test. Regarding the correlation between continuous variables, the Spearman rank test was used. Overall, survival was defined as the time from the initial NSCLC diagnosis to the date of death of any cause or the last follow-up if the patient was alive. Disease-free survival was determined as the duration from the initial NSCLC diagnosis to disease recurrence. The Kaplan–Meier method was used for survival estimates. Meanwhile, for multivariate survival analyses, the Cox proportional hazard model was used. A value of *p* < 0.05 was considered as statistically significant.

## 5. Conclusions

Our results support the theory that PD-L1^+^ lymphocytes are associated with the distribution of the CD4^+^ T cells, Foxp3^+^CD4^+^ T cells, and IL17A^+^CD4^+^ T cells, as well as the phenotypes of macrophages in a tumour microenvironment of resected NSCLC, and further research of these cell distributions in tumour islets and stroma could be a potential predictive and prognostic factor.

## Figures and Tables

**Figure 1 ijms-25-10827-f001:**
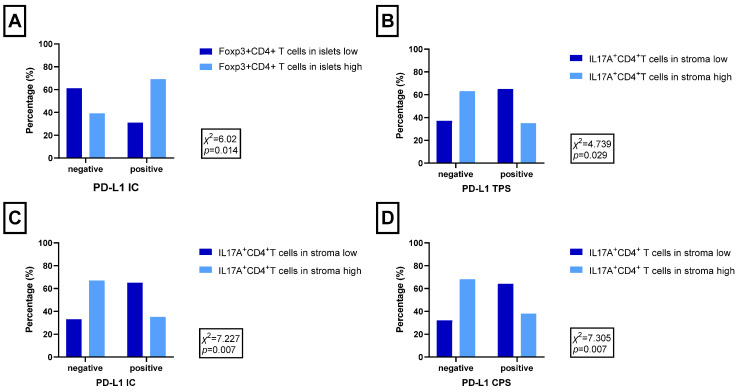
Immune cell distribution in TME depending on PD-L1 positivity: (**A**) Foxp3^+^CD4^+^ T cells in tumour islets and PD-L1 evaluated by IC. (**B**) IL-17A^+^CD4^+^ T cells in tumour stroma and PD-L1 evaluated by TPS, (**C**) IL-17A^+^CD4^+^ T cells in tumour stroma and PD-L1 evaluated by IC, and (**D**) IL-17A^+^CD4^+^ T cells in tumour stroma and PD-L1 evaluated by CPS.

**Figure 2 ijms-25-10827-f002:**
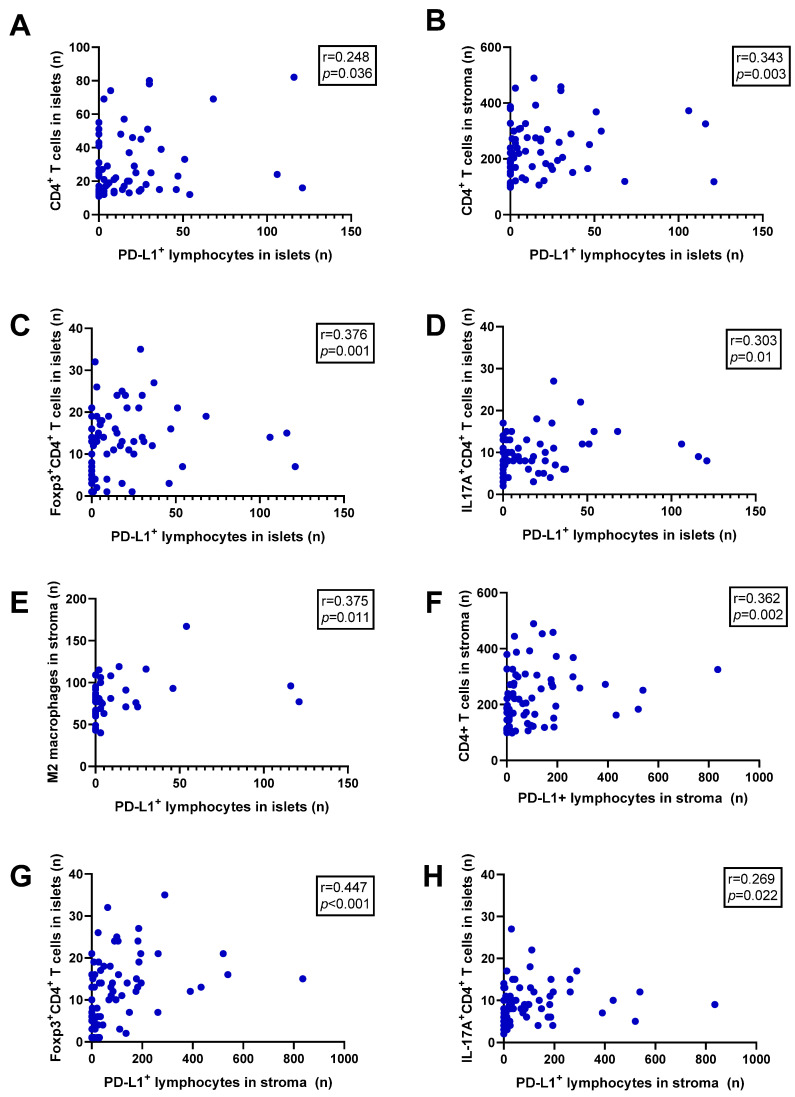
Correlation between PD-L1^+^ lymphocyte infiltration in tumour tissue (islets and stroma) and tumour-infiltrating immune cell distribution: (**A**) PD-L1^+^ lymphocytes in tumour islets and CD4^+^ T cells in tumour islets; (**B**) PD-L1^+^ lymphocytes in tumour islets and CD4^+^ T cells in tumour stroma; (**C**) PD-L1^+^ lymphocytes in tumour islets and Foxp3^+^CD4^+^ T cells in tumour islets; (**D**) PD-L1^+^ lymphocytes in tumour islets and IL-17A^+^CD4^+^ T cells in tumour islets; (**E**) PD-L1^+^ lymphocytes in tumour islets and M2 macrophages in tumour stroma; (**F**) PD-L1^+^ lymphocytes in tumour stroma and CD4^+^ T cells in tumour stroma; (**G**) PD-L1^+^ lymphocytes in tumour stroma and Foxp3^+^CD4^+^ T cells in tumour islets; (**H**) PD-L1^+^ lymphocytes in tumour stroma and IL-17A^+^CD4^+^ T cells in tumour islets.

**Figure 3 ijms-25-10827-f003:**
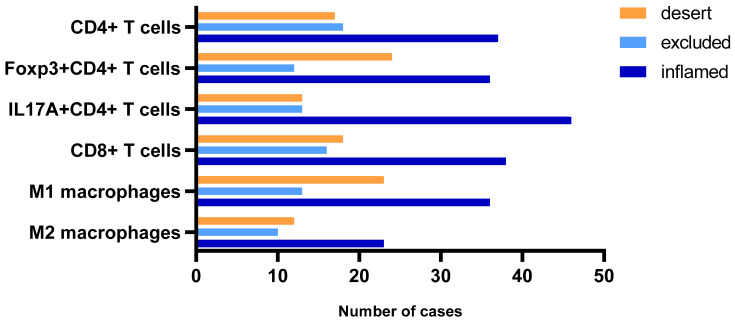
Tumour-immune phenotype depending on infiltrating immune cell type.

**Figure 4 ijms-25-10827-f004:**
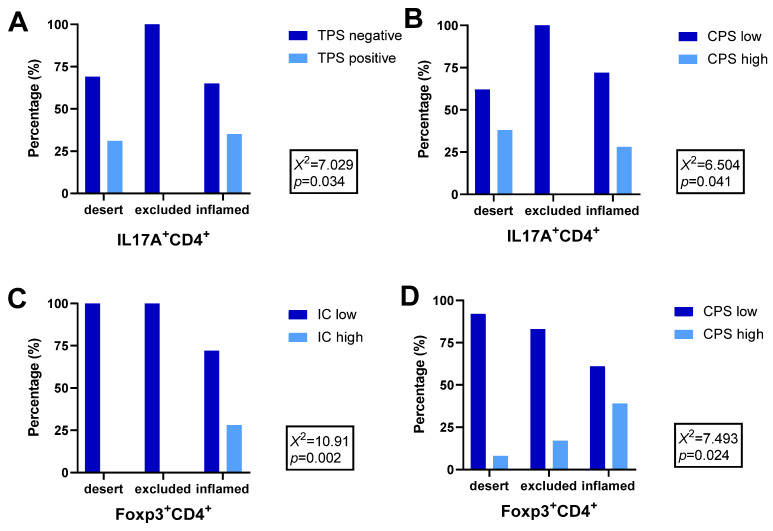
Tumour-immune phenotype depending on the level of PD-L1 expression: (**A**) IL17A^+^CD4^+^-immune phenotype depending on PD-L1 expression evaluated by TPS (negative or positive); (**B**) IL17A^+^CD4^+^-immune phenotype depending on PD-L1 expression evaluated by CPS (low or high), (**C**) Foxp3^+^CD4^+^-immune phenotype depending on PD-L1 expression evaluated by IC (low or high); (**D**) Foxp3^+^CD4^+^-immune phenotype depending on PD-L1 expression evaluated by CPS (low or high).

**Figure 5 ijms-25-10827-f005:**
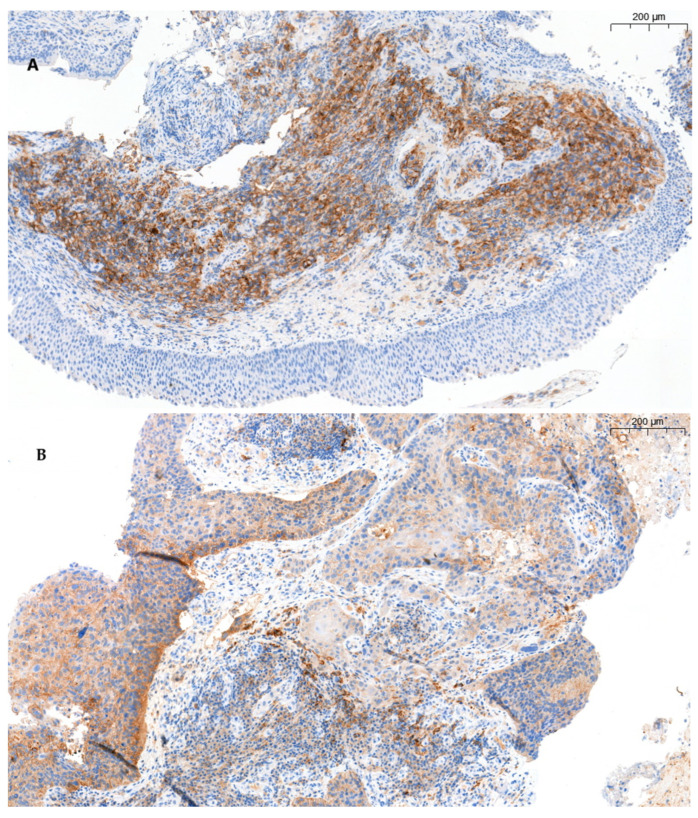
PD-L1 positivity in tumour cells and infiltrating lymphocytes: (**A**) PD-L1 positivity in tumour cells. (**B**) PD-L1 positivity in tumour cells and surrounding immune cells (lymphocytes). PD-L1—programmed death ligand 1.

**Table 1 ijms-25-10827-t001:** Characteristics of the study population.

Baseline Characteristics	*n* (%)
Gender	
Male	58 (80.6)
Female	14 (19.4)
Age group	
<65 years	33 (45.8)
≥65 years	39 (54.2)
Smoking status	
Non-smokers	13 (18.1)
Current or former smokers	59 (81.9)
COPD	
Absent	50 (69.4)
Present	22 (30.6)
Histological NSCLC type	
Adenocarcinoma	36 (50)
Squamous cell carcinoma	30 (41.7)
Large cell carcinoma	6 (8.3)
Differentiation	
Well—moderate	39 (54.2)
Poor—undifferentiated	33 (45.8)
NSCLC stage	
IA-IB	20 (27.8)
IIA-IIB	24 (33.3)
IIIA-IIIB	28 (38.9)
pT status	
pT1	12 (16.7)
pT2	41 (56.9)
pT3–4	19 (26.4)
Lymph node status	
pN0	31 (43.1)
pN1	23 (31.9)
pN2	16 (22.2)
pN3	2 (2.8)
Adjuvant therapy	
Given	31 (43.1)
Not given	41 (56.9)

**Table 2 ijms-25-10827-t002:** Association between PD-L1^+^ lymphocyte distribution and patient’s characteristics.

	PD-L1^+^ Lymphocytes					
	Islets	*p*	Stroma	*p*	Total	*p*
Gender						
Male	7.5 (0–121)	0.326	41 (0–836)	0.207	53 (0–952)	0.160
Female	1.5 (0–51)		18 (0–539)		19.5 (0–586)	
Age group						
<65 years	2 (0–106)	0.036	28 (0–390)	0.221	32 (0–391)	0.157
≥65 years	14 (0–121)		68 (0–836)		89 (0–952)	
Smoking status						
Nonsmokers	0 (0–68)	0.1	8 (0–187)	0.043	8 (0–255)	0.038
Former smokers and smokers	9 (0–121)		44 (0–836)		60 (0–952)	
COPD						
Absent	3 (0–121)	0.240	36.5 (0–521)	0.722	39 (0–542)	0.695
Present	14.5 (0–116)		60 (0–836)		82.5 (0–952)	
Histological NSCLC type						
Adenocarcinoma	4.5 (0–121)	0.596	29 (0–196)	0.104	36 (0–302)	0.114
Squamous cell carcinoma	7.5 (0–116)		88.5 (0–836)		99.5 (0–952)	
Differentiation						
Well-moderate	6 (0–116)	0.751	35 (0–836)	0.946	40 (0–952)	0.977
Poor	3 (0–121)		76 (0–539)		90 (0–586)	
NSCLC stage						
IA-IB	11.5 (0–28)	0.395	115 (38–433)	0.123	126.5 (38–458)	0.132
IIA-IIB	10 (0–106)		28 (0–262)		38 (0–316)	
IIIA-IIIB	4 (0–121)		87 (0–836)		103 (0–952)	
pT status						
pT1	11 (0–30)	0.667	86 (13–433)	0.042	97 (31–458)	0.063
pT2	3 (1–121)		25 (0–836)		34 (0–952)	
pT3–4	4 (0–47)		76 (0–539)		105 (0–586)	
Lymph node status						
Negative (pN0)	9 (0–106)	0.573	44 (0–539)	0.412	53 (0–586)	0.393
Positive (pN1–N3)	3 (0–121)		30 (0–836)		34 (0–952)	

The number of PD-L1-positive lymphocytes represents the median (range) per ten high-power fields in tumour tissue: islets, stroma and total—islets and stroma together. *p* values are from the Mann-Whitney U test and the Kruskal–Wallis test.

**Table 3 ijms-25-10827-t003:** Association between the infiltration levels of PD-L1^+^ lymphocytes and tumour-infiltrating immune cells.

	PD-L1^+^ Lymphocytes					
	Islets			Stroma		
	Low	High	*p*	Low	High	*p*
CD8^+^ T cells in islets n (%)						
Low	20 (58.8)	14 (41.2)	0.101	19 (55.9)	15 (44.1)	0.345
High	15 (39.5)	23 (60.5)		17 (44.7)	21 (55.3)	
CD8^+^ T cells in stroma n (%)						
Low	17 (50)	17 (50)	0.824	15 (44.1)	19 (55.9)	0.345
high	18 (47.4)	20 (52.6)		21 (55.3)	17 (44.7)	
CD4^+^ T cells in islets n (%)						
low	21 (60)	14 (40)	0.06	22 (62.9)	13 (37.1)	0.034
high	14 (27.8)	23 (62.2)		14 (37.8)	23 (62.2)	
CD4^+^ T cells in stroma n (%)						
Low	22 (62.9)	13 (37.1)	0.06	21 (60)	14 (40)	0.034
High	14 (37.8)	23 (62.2)		14 (37.8)	23 (62.2)	
Foxp3^+^CD4^+^ T cells in islets n (%)						
Low	23 (63.9)	13 (36.1)	0.009	24 (66.7)	12 (33.3)	0.005
High	12 (33.3)	24 (66.7)		12 (33.3)	24 (66.7)	
Foxp3^+^CD4^+^ T cells in stroma n (%)						
Low	22 (61.1)	14 (38.9)	0.034	21 (58.3)	15 (41.7)	0.157
High	13 (36.1)	23 (63.9)		15 (41.7)	21 (58.3)	
IL-17A^+^CD4^+^ cells in islets n (%)						
Low	17 (65.4)	9 (34.6)	0.032	17 (65.4)	9 (34.6)	0.05
High	18 (39.1)	28 (60.9)		19 (41.3)	27 (58.7)	
IL-17A^+^CD4^+^ cells in stroma n (%)						
Low	14 (43.8)	18 (56.3)	0.460	13 (40.6)	19 (59.4)	0.155
High	21 (52.5)	19 (47.5)		23 (57.5)	17 (42.5)	
M1 macrophages in islets n (%)						
Low	19 (52.8)	17 (47.2)	0.479	18 (50)	18 (50)	1.0
High	16 (44.4)	20 (55.5)		18 (50)	18 (50)	
M1 macrophages in stroma n (%)						
Low	14 (40)	21 (60)	0.155	13 (37.1)	22 (62.9)	0.034
High	21 (56.8)	16 (43.2)		23 (62.2)	14 (37.8)	
M2 macrophages in islets n (%)						
Low	16 (72.7)	6 (27.3)	0.586	16 (72.7)	6 (27.3)	0.586
High	15 (65.2)	8 (34.8)		15 (65.2)	8 (34.8)	
M2 macrophages in stroma n (%)						
Low	15 (65.2)	8 (34.8)	0.322	15 (65.2)	8 (34.8)	0.731
High	15 (62.5)	9 (37.5)		16 (66.7)	8 (33.3)	

*p*-values are from the chi-square (χ^2^) test.

**Table 4 ijms-25-10827-t004:** Association between PD-L1^+^ lymphocyte infiltration level and immune phenotypes (based on T cell infiltration).

	PD-L1^+^ Lymphocytes in Islets			PD-L1^+^ Lymphocytes in Stroma		
	Low	High	*p*	Low	High	*p*
CD8^+^ n (%)						
Desert	9 (50)	9 (50)	0.15	7 (38.9)	11 (61.1)	0.075
Excluded	11 (68.8)	5 (31.3)		12 (75)	4 (25)	
Inflamed	15 (39.5)	23 (60.5)		17 (44.7)	21 (55.3)	
CD4^+^ n (%)						
Desert	11 (64.7)	6 (35.3)	0.147	12 (70.6)	5 (29.4)	0.066
Excluded	10 (55.6)	8 (44.4)		10 (55.6)	8 (44.4)	
Inflamed	14 (37.8)	23 (62.2)		14 (37.8)	23 (62.2)	
Foxp3^+^CD4^+^ n (%)						
Desert	17 (70.8)	7 (29.2)	0.017	17 (70.8)	7 (29.2)	0.013
Excluded	6 (50)	6 (50)		7 (58.3)	5 (41.7)	
Inflamed	12 (33.3)	24 (66.7)		12 (33.3)	24 (66.7)	
IL17A^+^CD4^+^ n (%)						
Desert	7 (53.8)	6 (46.2)	0.043	6 (45.2)	7 (53.8)	0.022
Excluded	10 (76.9)	3 (23.1)		11 (84.6)	2 (15.4)	
Inflamed	18 (39.1)	28 (60.9)		19 (41.3)	27 (58.7)	

*p* values are from the chi-square (χ^2^) test.

**Table 5 ijms-25-10827-t005:** Association between PD-L1^+^ lymphocyte infiltration level and immune phenotypes (based on macrophage infiltration).

	PD-L1^+^ Lymphocytes in Islets			PD-L1^+^ Lymphocytes in Stroma		
	Low	High	*p*	Low	High	*p*
M1 macrophages n (%)						
Desert	10 (43.5)	13 (56.5)	0.295	9 (39.1)	14 (60.9)	0.24
Excluded	9 (69.2)	4 (30.8)		9 (69.2)	4 (30.8)	
Inflamed	16 (44.4)	20 (55.6)		18 (50)	18 (50)	
M2 macrophages n (%)						
Desert	10 (83.3)	2 (16.7)	0.481	10 (83.3)	2 (16.7)	0.481
Excluded	6 (60)	4 (40)		6 (60)	4 (40)	
Inflamed	15 (65.2)	8 (34.8)		15 (65.2)	8 (34.8)	

*p*-values are from the chi-square (χ^2^) test.

## Data Availability

Data are contained within the article or the Appendix A.

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
