# Peer review of "PD-L1+ Lymphocytes Are Associated with CD4+, Foxp3+CD4+, IL17+CD4+ T Cells and Subtypes of Macrophages in Resected Early-Stage Non-Small Cell Lung Cancer"

_ijms, 2024, doi:10.3390/ijms251910827_

Round 1

Reviewer 1 Report

Comments and Suggestions for Authors

The authors found that PD-L1+ lymphocytes are associated with the distribution of CD4+ T cells, Foxp3+CD4+ T cells, and IL17A+CD4+ T cells, as well as the phenotypes of macrophages in the tumor microenvironment of resected non-small cell lung cancer (NSCLC). Further research into the distribution of these cells in tumor islets and stroma could serve as potential predictive and prognostic factors. This is an interesting article, and I have a few suggestions:

  1. As the authors described, the included patients underwent surgery between September 2012 and February 2015. However, considering the current situation, I recommend using at least the 8th edition of the TNM staging system.
  2. Some patients received adjuvant chemotherapy or radiotherapy. Could neoadjuvant therapy have impacted the pathological status at surgery for these patients, preventing an accurate reflection of their initial immune status?
  3. Please clarify that the sampled lesions were all from pulmonary lesions rather than surgically resected metastatic lymph nodes.
  4. Two patients were diagnosed as PN3. I am curious about how N3 status was determined for these patients.
  5. Since the patients included in this study were in early to mid-stage disease, it is suggested to mention that similar studies need to be validated in late-stage NSCLC patients.

Author Response

Thank You for comments, suggestions and insights, that will help us in our further research.

1. As the authors described, the included patients underwent surgery between September 2012 and February 2015. However, considering the current situation, I recommend using at least the 8th edition of the TNM staging system.

Thank you for the suggestion, however at that time the 7th edition of the TNM Classification of Malignant Tumours was used for lung tumour staging and the decision was made to leave the same classification.

2. Some patients received adjuvant chemotherapy or radiotherapy. Could neoadjuvant therapy have impacted the pathological status at surgery for these patients, preventing an accurate reflection of their initial immune status?

This study patients got only adjuvant chemotherapy and/or radiotherapy, but theoretically neoadjuvant treatment could affect immune response and further researches are required.

3. Please clarify that the sampled lesions were all from pulmonary lesions rather than surgically resected metastatic lymph nodes.

Yes, the sample lesions were all from pulmonary lesions and it was specified in study population part, line 83-84.

4. Two patients were diagnosed as PN3. I am curious about how N3 status was determined for these patients.

PN3 was diagnosed only after resection according to pathological report. Meanwhile, clinical staging demonstrated lover stage.

5. Since the patients included in this study were in early to mid-stage disease, it is suggested to mention that similar studies need to be validated in late-stage NSCLC patients.

We strongly agree that further studies need to be validated not only in early stage, but also in late-stage NSCLC patients.

Reviewer 2 Report

Comments and Suggestions for Authors

In this manuscript, the authors reveal that, within the tumor microenvironment (TME), PD-L1+ lymphocytes are associated with the distribution of CD4+ T cells, Foxp3+CD4+ T cells, IL-17A+CD4+ T cells, and M2 macrophages. These data highlight the distribution of various tumor-infiltrating immune cells, providing a more in-depth understanding of the relationship between PD-L1 expression and immune cell types in the TME. 

The authors defined PD-L1 expression using immunohistochemistry (IHC). However, no IHC images are shown in the figures. Could you please include the IHC results? Additionally, for the thick tissue sections, could the authors provide representative images showing PD-L1+, CD4+, CD8+ cells, and macrophages?

In Figure 1, the authors show the distribution of immune cells in the TME. However, the histogram plot is unclear. Is it representing just one data point? Could the authors include all the original data points?

In Figure 2, the authors found significant correlations between PD-L1+ lymphocytes and CD4+ T cells, IL-17A+CD4+ T cells, Foxp3+CD4+ T cells, and M2 macrophages. However, the r-value is less than 0.5, which suggests a moderate correlation. Could the authors provide more specific data to support this conclusion?

Comments on the Quality of English Language

Minor editing English language is required.

Author Response

Thank You for comments, suggestions and insights, that helped us to improve our manuscript.

- The authors defined PD-L1 expression using immunohistochemistry (IHC). However, no IHC images are shown in the figures. Could you please include the IHC results? Additionally, for the thick tissue sections, could the authors provide representative images showing PD-L1+, CD4+, CD8+ cells, and macrophages?

The IHC results of PD-L1 staining were added to the manuscript – line 148-149, Figure 1. Meanwhile the IHC results of the distribution of CD4+ T cells, Foxp3+CD4+ T cells, IL-17A+CD4+ T cells, and M2 macrophages were already published in previous articles, mentioned in study population part: line 84-87.

- In Figure 1, the authors show the distribution of immune cells in the TME. However, the histogram plot is unclear. Is it representing just one data point? Could the authors include all the original data points?

In this study immune cells distribution were assessed only in one point – resected lung tumour tissue.

- In Figure 2, the authors found significant correlations between PD-L1+ lymphocytes and CD4+ T cells, IL-17A+CD4+ T cells, Foxp3+CD4+ T cells, and M2 macrophages. However, the r-value is less than 0.5, which suggests a moderate correlation. Could the authors provide more specific data to support this conclusion?

These results reached the significant level due to p value < 0,05. However, R value ranged from week to moderate correlation and this was additionally mentioned in line 240. Our results could demonstrate only week to moderate correlation due to small sample size. However, similar results were found using other statistical method represented in table 2.

Reviewer 3 Report

Comments and Suggestions for Authors

1. Some potential confounders (patients' age, sex, smoking status, and comorbidities (e.g., COPD)) may also affect immune cell infiltration and PD-L1 expression, whether or not they were appropriately statistically adjusted.

2. The roles of Foxp3+CD4+ T cells and IL17A+CD4+ T cells may be different at different tumor stages, and the effect of immune phenotype changes on PD-L1 expression needs to be further explored.

3. For results that did not reach the significance level, it is recommended to discuss the possible reasons in more detail, such as insufficient sample size, complexity of immune phenotype, or limitations of cell distribution, to avoid simple interpretation of results.

4. The results showed that PD-L1+ lymphocyte distribution was not significantly associated with survival, but possible influencing factors were not further explored. It is recommended to discuss the role of other clinical and biomarkers in survival to supplement the understanding of PD-L1.

Author Response

Thank You for comments, suggestions and insights, that will help us in our further research.

  1. Some potential confounders (patients' age, sex, smoking status, and comorbidities (e.g., COPD)) may also affect immune cell infiltration and PD-L1 expression, whether or not they were appropriately statistically adjusted.

When analysing associations between PD-L1+ lymphocytes and clinicopathological factors associations were found only with patient’s age and smoking status, whereas there were no associations between PD-L1+ lymphocytes and sex, COPD, stage, histological NSCLC type and others, so for simple statistical tests, such as Mann-Whitney, Kruskal–Wallis, Chi-square (χ2) test and others, no statistical adjustment was performed.

  1. The roles of Foxp3+CD4+ T cells and IL17A+CD4+ T cells may be different at different tumor stages, and the effect of immune phenotype changes on PD-L1 expression needs to be further explored.

In our study significant association between PD-L1+ lymphocytes, CD4+ T cell subsets ant tumour stage was not found, but in this case the main part of study population were stage I-II. Theoretically, including advanced and metastatic NSCLC tumours, the role of infiltrating immune cells could be different, so the effect of immune phenotype changes on PD-L1 expression needs to be further explored.

  1. For results that did not reach the significance level, it is recommended to discuss the possible reasons in more detail, such as insufficient sample size, complexity of immune phenotype, or limitations of cell distribution, to avoid simple interpretation of results.

We presented small sample size as one of the main limitations of our research in line 501, however we agree, that complexity of the immune phenotype, or limitations of cell distribution could also affect the results. This is attempted to be highlighted in the discussion section.

  1. The results showed that PD-L1+ lymphocyte distribution was not significantly associated with survival, but possible influencing factors were not further explored. It is recommended to discuss the role of other clinical and biomarkers in survival to supplement the understanding of PD-L1.

While the aim of this study was to evaluate PD-L1+ lymphocytes distribution in TME in the context of tumour immune phenotype in early-stage resected NSCLC patients, we evaluated the association between PD-L1+ lymphocytes distribution and survival. Associations between survival and other immune cells, including CD8+ T cells, CD4+ T cells, as well as subsets of macrophages were discussed in previous articles, mentioned in line 84-87. While other articles represented, that PD-L1+ lymphocytes alone were not a prognostic or predictive factor, but together with other immune cells were associated with survival, we also analysed weather PD-L1+ lymphocytes together with other immune cells were associated with survival, however statistically significant associations were not obtained in this case either.